# Prevention of Musculoskeletal Diseases and Pain among Dental Professionals through Ergonomic Interventions: A Systematic Literature Review

**DOI:** 10.3390/ijerph17103482

**Published:** 2020-05-16

**Authors:** Janna Lietz, Nazan Ulusoy, Albert Nienhaus

**Affiliations:** 1Competence Center for Epidemiology and Health Services Research for Healthcare Professionals (CVcare), University Medical Center Hamburg-Eppendorf (UKE), Martinistrasse 52, 20246 Hamburg, Germany; janna.lietz@gmx.de(J.L.); albert.nienhaus@bgw-online.de (A.N.); 2Department of Occupational Medicine, Hazardous Substances and Public Health (AGG), Institution for Statutory Accident Insurance and Prevention in the Health and Welfare Services (BGW), Pappelallee 33/35/37, 22089 Hamburg, Germany

**Keywords:** oral health, primary prevention, ergonomics, musculoskeletal diseases, systematic reviews as topic

## Abstract

Musculoskeletal diseases and pain (MSDs) are prevalent among dental professionals. They cause a growing inability to work and premature leaving of the occupation. Thus, the objective of this review was to summarize the evidence of ergonomic interventions for the prevention of MSDs among dental professionals. This review was conducted using Preferred Reporting Items for Systematic Reviews and Meta-Analyses (PRISMA) guidelines. The literature search was carried out in May 2018, with an update in April 2019. Scientific databases such as MEDLINE, CINAHL, PubMed and Web of Science as well as reference lists of the included studies were used. Relevant data were extracted from the studies and summarized. The quality assessment was performed using a validated standardized instrument. Eleven studies were included in this review, of which four are of high quality. Eight studies focused on setting prevention strategies. Of those, in five studies, magnification loupes or prismatic spectacles were the subject of ergonomic interventions. Further subjects were the dental chair (*n* = 2) and dental instruments (*n* = 1). Three studies evaluated ergonomic training. In all studies, the ergonomic interventions had positive effects on the study outcome. Several ergonomic interventions to prevent MSDs among dental professionals were found to exert a positive effect on the prevalence of MSDs or working posture. This systematic review adds current evidence for the use of prismatic spectacles in order to prevent MSDs among dental professionals. Further intervention studies about the role of ergonomics for the prevention of MSDs among dental professionals are warranted.

## 1. Introduction

Musculoskeletal diseases and pain (MSDs) show a high prevalence among dental professionals. Our last systematic review yielded an overall pooled annual prevalence of 78 percent. MSDs were most common in the neck (58.5%), lower back (56.4%), shoulder (43.1%) and upper back (41.1%) [1]. Often, MSDs are described as a group of diseases and complaints that have impacts on various structures of the musculoskeletal system of humans. These comprise, for example, joints, muscles, bones, nerves, blood vessels, ligaments, tendons, and supporting structures like intervertebral discs [2,3,4]. MSDs can arise from one or several injuries and result in pain or sensory disturbances in several body regions. They can become a temporary or chronic illness—The latter is more common, representing 40% of all chronic diseases [3,4]. Several studies have found that MSDs often lead to a growing inability to work, sick leave, a poorer quality of work, decreased job satisfaction, work-related accidents, and premature leaving of the occupation [2,3,5,6]. Furthermore, MSDs can cause high health care expenditures for the medical treatments of the diseased. The health and economic burdens of MSDs are considerable [6].

To be free of serious MSDs is of high importance, especially for dental professionals, as dental care is a physically and mentally demanding occupation. During their work, dental professionals have to carry out precise movements with their hands, adopt awkward working postures, use vibrating dental instruments, and do administrative work and repetitive monotonous tasks over a long time [1,7]. As a consequence, the prevention of MSDs is particularly important in dental care in order to decrease the mentioned risk factors, frequency of severe symptoms, high prevalence rates, and, in the long-term, promote a good physical and mental health status among dental professionals. There are some systematic literature reviews focusing on ergonomic interventions to prevent MSDs among dental professionals [8,9,10]. The last one was published by Roll and colleagues [10] in 2019. Our systematic review presents the most current state of the research on the prevention of MSDs among dental professionals through ergonomic interventions, as we included studies not then known to Roll and colleagues [10]. In addition, by using different eligibility criteria, we included studies not described in the previous reviews.

Overall, the objective of this systematic review was to summarize the evidence of ergonomic interventions for the prevention of MSDs among dental professionals. The focus here is primarily on extrinsic factors such as ergonomic seating facilities or visual aids.

## 2. Methods

This literature review was conducted systematically in accordance with the Preferred Reporting Items for Systematic Reviews and Meta-Analyses (PRISMA) guidelines [11]. The related study protocol was written in line with the Preferred Reporting Items for Systematic Review and Meta-Analysis Protocols (PRISMA-P) statement [12]. It is available in English and describes the planned research methods for this review in more detail. The protocol can be obtained from the corresponding author on request.

Neither an ethics committee approval nor informed consent were necessary for this systematic review of published literature. There was no contact with real study participants at any time.

### 2.1. Eligibility Criteria

For the screening and eligibility assessment of identified studies in databases and reference lists, various criteria were developed in accordance with the Population, Intervention, Control group, Outcome and Study design (PICOS) scheme (Table 1) [13]. Additional criteria specified by the authors were also considered like the language, publication status and date.

Studies were included in the review if the study population comprised dental professionals working in dental care facilities as dental practices, dental clinics/hospitals or dental schools. Dental professionals consisted of, for instance, dentists, orthodontists, dental assistants/hygienists/technicians and dental students. Moreover, studies were considered if the performed *intervention* focused on ergonomic design options for work in dentistry. Such ergonomic interventions can examine the effect of, for example, magnification loupes, prismatic spectacles, dental instruments, dental chairs or lighting on the physical and mental health status of dental professionals. Interventions were only considered appropriate if they lasted for at least two days. If the duration of the intervention is too short, the effectiveness of the measure is difficult to prove and questionable. Furthermore, the authors only selected studies in which suitable *control groups* were included as a comparison for the intervention groups to emphasize the effect of the measure itself. Studies were also included if the subjects themselves represented both the intervention and the control group (own controls). The *outcome* measures should be related to MSDs on the basis of prevalence or symptoms. Studies were also included if the outcome measures were related to working postures as these likely link to MSDs. We considered all possible effects of the ergonomic intervention on the study outcome. This can be a positive or negative change in MSDs or working postures as well as no change, respectively. We included all studies with these possible outcomes. Regarding the *study design* intervention or evaluation studies, randomized controlled trials (RCTs) and observational studies (e.g., cohort studies) were included, once the effect of the ergonomic intervention had been clearly analyzed. Furthermore, the authors only considered studies written in English. The studies had to be published ideally in peer-reviewed journals and accessible as full texts with preceding abstracts. Finally, studies were included if they were published between January 2008 and April 2019. In order to present current and relevant results in this literature review, we have decided to include only studies from the last 10 years. In April 2019, an update search was performed in order to verify whether new relevant studies had been published since May 2018.

### 2.2. Information Sources and Search Strategy

The systematic literature search was performed in May 2018. It was applied to the scientific databases MEDLINE, CINAHL, PubMed and Web of Science. Furthermore, reference lists of the included studies and relevant review articles were examined to identify further sources. Other scientific experts of this study topic were contacted by email or in person to receive additional information about current publications and research projects.

The following search terms and Medical Subject Headings (MeSH) were used to search in all included databases, for example:

Dent* OR dental personnel/ professionals OR oral health

AND

Ergonomics/human engineering OR intervention OR (primary) prevention

AND

Musculoskeletal diseases OR musculoskeletal pain.

A detailed description of the general search strategy is provided in Appendix A. The strategy and search terms were adapted to the setup of the individual scientific database.

A systematic update search in April 2019 revealed one further study fulfilling the eligibility criteria of this review [14].

### 2.3. Literature Screening

The literature screening and corresponding eligibility assessment of the studies found were carried out independently by two authors (J.L. and N.U.). The screening process consisted of a title and abstract screening as well as a full-text screening. For this purpose, a standardized screening instrument was developed based on the PICOS criteria. If a study met all the predefined eligibility criteria, it was included in the review. Disagreements between the two authors were resolved by discussion. J.L. and N.U. ultimately agreed on all the included studies.

### 2.4. Data Collection

The data collection for the included studies was performed by J.L. and N.U. independently. A standardized data extraction tool was developed for collecting and summarizing information on several study characteristics (e.g., study design, study region, setting and study population) and study results (e.g., sample size and the effect of the intervention). The extraction tool comprised 16 relevant items, including the PICOS criteria of this study. In the case of uncertainty, a discussion was held between the authors. J.L. extracted the detailed data using Microsoft Excel 2013 (Microsoft, Redmond, DC, USA) spreadsheets. N.U. checked the accuracy of the data. If possible, a calculation of missing values was conducted by J.L. Some study’s corresponding authors were contacted by email to obtain more information on the presented results or missing data.

### 2.5. Quality Assessment

Following the screening and data extraction, the included studies were assessed in terms of study quality. The assessment was performed by two authors (J.L. and N.U.) independently. Differing results were discussed and resolved among the authors. For this quality assessment, a standardized fully validated instrument by the scientists Downs and Black [15] was used. The instrument was originally developed for the assessment of the methodological quality both of randomized and non-randomized studies of health care interventions. The validation of the instrument showed very good results (e.g., reliability: r = 0.88) [15]. It comprised 27 items that were categorized by five quality criteria. An overview of the individual items is provided in Appendix A. The items (e.g., “are the main findings of the study clearly described?”) were to be answered with “yes” (1 point), “no” (0 points), or “unable to determine” (0 points). There were two exceptions; Item 5 was to be answered with “yes” (2 points), “partially” (1 point), or “no” (0 points), and Item 27 was to be answered on a scale of 1 to 5 possible points, representing the quality of the power of the respective study. The study quality was finally assessed by adding up the points. This yielded a scale from 0 to 32 points. Studies with a score from 32 to 22 points were considered of high quality; studies with a score from 21 to 11 points, of moderate quality; and studies with a score from 10 to 0 points, of low quality.

### 2.6. Statistical Analysis and Data Synthesis 

Following the quality assessment, the included studies underwent a descriptive analysis and a narrative summary of the key results was prepared. Differences and similarities in methods and results were emphasized and described. These elaborations formed the basis for the decision as to whether a meta-analysis could be conducted for this systematic literature review.

Eventually, no meta-analysis was performed, as out of 11 studies, only one study was clearly suitable for inclusion in a meta-analysis [16]. When looking more closely at the other studies, it becomes clear that they do not fulfill the criteria for a meta-analysis as they do not have appropriate raw data that could be used. A relevant barrier was also that the studies were, overall, very heterogeneous. The survey instruments, outcome measures and subjects of intervention varied widely.

## 3. Results

### 3.1. Study Selection

The database search yielded 216 titles (Figure 1). Through reference searching, 19 additional studies were identified. Through update searching, six additional studies were found. After the removal of duplicates, 190 titles remained. Of these, 148 studies were excluded after the title and abstract screening, as they did not fulfill the predefined eligibility criteria. Of the remaining 42 studies that were subjected to the full-text screening, 25 did not meet the eligibility criteria. Six sources were excluded afterwards as they reported rehabilitation measures and not ergonomic interventions. The main reasons for exclusion from this review were a different study topic (e.g., interventions related to the rehabilitation, prevalence and non-occupational risk factors of MSDs), study population (e.g., other occupational groups) or study design (e.g., a review, or descriptive cross-sectional study). In the end, 11 studies were considered suitable to be included in this literature review. They consisted of two randomized controlled trials (RCTs), six intervention studies, two evaluation studies and one cohort study.

### 3.2. Study Characteristics

All the included sources were scientific studies analyzing the effect of ergonomic interventions to prevent MSDs among dental professionals. They were published in the English language between 2008 and 2018, with 2016 having the most studies issued per year (*n* = 4; Table 2).

The studies were conducted in different countries across four continents. Four studies came from Asia (India and Iran), three from Europe (Italy and Sweden) and two each from North America (Canada and United States) and from Oceania (Australia). Almost half of the studies (*n* = 5) took place in dental schools/universities and dental hospitals/clinics. One study was carried out in a dental practice. A variety of dental professionals represented the study population of the included studies, for example dentists (*n* = 6), dental (hygiene) students (*n* = 5) and dental hygienists (*n* = 5). In some studies, several dental professionals were included. The sample size ranged from 29 to 564 subjects, with an average of 106 subjects. The relatively high number of study participants (*n* = 564) comes from a longitudinal cohort study [23]. In five studies, separate intervention and control groups were involved, and in six studies, each subject represented both the intervention and control group. The number of each group of the investigation is shown in Table 2. In Rempel and colleagues [25] both groups were intervention groups with their own controls. In most of the studies (*n* = 6), the symptoms of MSDs were the related study outcome, followed by working posture (*n* = 4) and the prevalence of MSDs (*n* = 1). The direction of the study outcomes (effect of the ergonomic intervention) in each included study is described in Section 3.3. *Ergonomic Interventions.* Several ergonomic measures were used in the studies to analyze their effects on the prevalence/symptoms of MSDs or working posture. Based on the 27 quality criteria according to Downs and Black [15], four studies (36%) were classified as high quality (32–22 points) and seven studies (64%) as moderate quality (21–11 points), with an average of 19.6 points. The most common reasons for a moderate methodological quality were weaknesses in the study design (no randomization or no blinding), data analysis (no adjustment for confounding factors or no calculation of power) and missing information about losses to follow up(s).

### 3.3. Ergonomic Interventions

The studies included in this systematic review focused on ergonomic interventions to prevent MSDs or to improve working postures among dental professionals. The studies used a variety of ergonomic interventions to analyze their effects on the frequency and severity of MSDs and the working posture of dental professionals (Figure 2).

In five studies (46%), magnification loupes [20,21,24] or prismatic spectacles [22,23] were the subject of ergonomic interventions. Further subjects were the ergonomic dental chair (*n* = 2, 18%) [17,19] and dental instruments (*n* = 1, 9%) [25]. Three studies (27%) evaluated ergonomic training that aimed at reducing MSDs among dental professionals [14,16,18]. Dable and colleagues [17] primarily investigated the effect of an ergonomic dental chair on the working posture of dental professionals. The authors added magnification loupes and a lecture in ergonomics to their investigation. Additionally, Lindegård and colleagues [22] supplemented lectures in ergonomics to their investigation of prismatic spectacles.

In the following section, the ergonomic interventions of the included studies are described in more detail including an analysis of the most relevant effects of the interventions on the individual study outcomes.

Eight studies (73%) focused on setting prevention and three studies (27%), on behavioral prevention strategies while conducting ergonomic interventions to reduce MSDs or to improve working posture among dental professionals (Table 3). 

Regarding the survey instruments in the studies, standardized and validated questionnaires (e.g., the NMQ, DASH questionnaire and NPDS scale) (72.7%, *n* = 8) and questionnaires with self-developed questions (63.6%, *n* = 7) were most commonly used to assess the effect of the ergonomic intervention. Further instruments were photographs or videotapes (36.3%, *n* = 4), posture assessment instruments (e.g., RULA and PAI) (27.2%, *n* = 3) and physical examinations (27.2%, *n* = 3). The length of the ergonomic intervention ranged from one week to one year. In most of the studies (*n* = 7), the study participants experienced MSDs before enrolment in the study and its ergonomic intervention. In four studies, this information was missing. During the implementation of the ergonomic interventions, their impact on MSDs and working posture in the various body regions of dental professionals was investigated. In the studies, the most commonly analyzed body regions were the neck (72.7%, *n* = 8) and the shoulder (72.7%, *n* = 8). Other analyzed body regions were the arm (63.6%, *n* = 7), wrist (45.4%, *n* = 5), leg/thigh (45.4%, *n* = 5), back (36.3%, *n* = 4) and head (36.3%, *n* = 4). In all the studies, the ergonomic interventions had positive effects on the prevalence of MSDs or working posture among dental professionals.

#### 3.3.1. Ergonomic Dental Chair

Dable and colleagues [17] and Hallaj and colleagues [19] both investigated the influence of ergonomic dental chairs (without and with magnification loupes) on the working posture of dental professionals. Significantly lower RULA scores for the ergonomic dental chair with magnification (1.57 ± 0.50) as compared to conventional chairs without magnification (7.03 ± 0.49) were found by Dable and colleagues [17]. Consequently, the use of the ergonomic dental chair with magnification was more suitable and produced a better working posture than the use of the conventional chairs without magnification. Working posture significantly improved following the use of the ergonomic dental chair with magnification (p<0.01), and dental students reported fewer or no MSDs as they found the ergonomic dental chair more comfortable than the conventional chairs. The ergonomic dental chair could support the lumbar region and maintain the natural curvature of the lower back, and magnification could bring a clearer view nearer to the dental students [17]. Hallaj and colleagues [19] found similar results. In their study, the overall RULA score was 3.14 after the use of an ergonomic dental chair with arm support [19]. As a result, the use of the ergonomic dental chair led to positive changes in the working posture of dentists and put it almost in the correct ergonomic position. Further study results confirmed this; the combined bending and twisting of the back decreased by 13.8% following the intervention, the excessive bending up or down of the wrist decreased by 41.4% and the pressure on neck and shoulder during dental tasks decreased by 79.3%. Furthermore, dentists reported more comfort when using the arm support device. Working posture can further be improved by adjusting both the patient’s and dentist’s chairs to support the dentist’s neck [19].

#### 3.3.2. Magnification Loupes

Four included sources [17,20,21,24] examined the effect of magnification loupes on health-related outcomes (symptoms of MSDs and working posture) among dental professionals. In one study [20], at baseline, the DASH scores for dental hygienists (intervention group) were higher than for dental hygiene students (control group) (8.56 ± 9.64 vs. 4.99 ± 6.25); after using magnification loupes, this trend was reversed (5.17 ± 5.29 vs. 7.84 ± 8.73). Consequently, through the intervention, the DASH scores for dental hygienists decreased and for dental hygiene students increased. The use of magnification loupes significantly reduced the symptoms of MSDs among dental hygienists (*p* < 0.05). Therefore, the symptoms of MSDs improved in the intervention group and worsened in the control group, which emphasized the positive effect of magnification loupes on the symptoms of MSDs in this study [20]. Another study from Hayes and colleagues [21] revealed results that were similar but were less meaningful, with smaller effects. The authors found no change in mean NPDS scores between baseline and follow up for dental hygienists (intervention group) (14.00 ± 12.49 vs. 14.00 ± 11.05), while dental hygiene students (control group) reported an increase in perceived neck pain at follow up (14.97 ± 16.91 vs. 15.90 ± 13.54; *p* > 0.05). As a consequence, the use of magnification loupes did not create significant changes in neck pain for dental hygienists but a slightly positive effect on the symptoms of neck pain can be assumed [21]. Maillet and colleagues [24] found a correlation between the use of magnification loupes and the working posture of dental hygiene students. All the students wearing the magnification loupes showed significantly better ergonomic mean scores than all the students not wearing them (6.4 ± 2.61 vs. 10.8 ± 4.24, t = 6.66, df = 34, *p* < 0.000001). As a result, the use of magnification loupes significantly improved the working posture of dental hygiene students in both investigated groups (*p* < 0.001). The authors stated that an early introduction in the use of magnification loupes is more effective in improving working posture. The majority of the students were aware of the improved working posture, perceived an increase in the quality of their work and would wear magnification loupes regularly [24]. Finally, Dable and colleagues [17] (described above) also reported a significantly positive impact of magnification loupes on the working posture of dental students.

#### 3.3.3. Prismatic Spectacles

Lindegård and colleagues [22,23] investigated the influence of prismatic spectacles on health-related outcomes (working posture and the symptoms of MSDs) among dental professionals. In one study, the head and neck flexion was reduced at follow up in both groups, but more pronounced in the intervention group (received prismatic spectacles) than in the control group (did not receive prismatic spectacles) (8.7° vs. 3.6°, *p* < 0.01, and 8.2° vs. 3.3°, *p* < 0.05). Furthermore, there was a significant decrease (4 units) in the perceived exertion of the head and the neck in the intervention group; the decrease in the control group was 2 units [22]. Therefore, the use of prismatic spectacles caused significant positive changes in working posture and reduced complaints by dentists and dental hygienists for the head and neck regions. Eighty percent of the participants reported that the use of prismatic spectacles considerably facilitated their work [22]. Another study by Lindegård and colleagues [23] showed comparable results. The study revealed significant improvements regarding clinical diagnoses (*p* < 0.05), perceived exertion (*p* < 0.01), self-reported pain (*p* < 0.05) and self-rated work ability (*p* < 0.05) in the intervention group (used prismatic spectacles) as compared to the control group (did not receive prismatic spectacles). Consequently, the use of prismatic spectacles significantly improved symptoms and reduced the risk of MSDs in dental personnel. Study participants reported that using the prismatic spectacles simplified their dental work and strengthened their work ability. The greatest advantage was found during root-fillings and other vision-demanding tasks in constrained working positions. The spectacles enabled dental work in a more upright position with a less bent neck, which promoted an ergonomic working posture with a lower risk of MSDs [23].

#### 3.3.4. Dental Instruments

In one study [25], the impact of two different dental instruments on the symptoms of MSDs in dentists and dental hygienists was analyzed. The authors compared the use of a lightweight dental instrument with a wide diameter (Instrument I) with a heavy dental instrument with a narrow diameter (Instrument II). The unadjusted pain scores for the study participants who used Instrument I improved more than for the participants who used Instrument II for the wrist/hand (0.40 ± 0.11 vs. 0.14 ± 0.11, n. s.), arm (0.20 ± 0.09 vs. 0.06 ± 0.09, n. s.) and shoulder (0.51 ± 0.16 vs. 0.19 ± 0.15, *p* < 0.05) regions. After adjusting for confounders (e.g., age and occupation), the authors only found a significant difference between the two groups for the shoulder region (0.52 ± 0.17 vs. 0.19 ± 0.16, *p* < 0.05). As a result, the use of the lightweight dental instrument with a wide diameter was more suitable for dental work than the use of a heavy instrument with a narrow diameter, even if in both groups, the symptoms of MSDs improved. The improvements were greater among participants who used Instrument I. The use of this instrument significantly reduced the symptoms of shoulder pain and showed higher improvements regarding the number of nights awakening with finger numbness than Instrument II. Finally, the ratings regarding the usability of the two instruments revealed more positive results for the use of the lightweight instrument with a wide diameter [25].

#### 3.3.5. Training Course in Ergonomics

Three included studies [14,16,18] examined the association between participation in a training course in ergonomics and the frequency or severity of MSDs among dental professionals. Dehghan and colleagues [16] found that dentists in the intervention group had lower prevalence rates of MSDs for all body regions 3 and 6 months after the intervention than dentists in the control group. For instance, the prevalence of knee pain was 24% vs. 36% (*p* < 0.01); of shoulder pain, 44% vs. 80% (*p* < 0.05); and of neck pain, 62% vs. 84% (*p* < 0.01) 6 months after the program. Moreover, in the intervention group, the prevalence rates of MSDs decreased over time for all body regions, and in the control group, only for the back region. Consequently, the ergonomic intervention program had a positive effect by significantly reducing the prevalence of MSDs in dentists. Knowledge about ergonomics and workplace modification in dental care can improve experiences of MSDs. Almost all participants (98%) agreed with the ergonomic intervention program, experienced benefits, had significantly fewer MSDs after the intervention and were able to improve their workplace [16]. Farrokhnia and colleagues [18] and Koni and colleagues [14] found similar results in their studies. In one study [18], the means for MSDs for the neck (10.97 ± 20.44 vs. 7.91 ± 17.01, *p* < 0.01), right shoulder (8.85 ± 19.76 vs. 5.24 ± 13.51, *p* < 0.01), left shoulder (5.80 ± 17.21 vs. 2.95 ± 9.33, *p* < 0.01), upper back (6.92 ± 17.59 vs. 4.53 ± 14.35, *p* < 0.01) and right wrist (5.12 ± 13.35 vs. 3.81 ± 12.96, *p* < 0.05) regions were significantly decreased at follow up. Before the intervention, 87% of dentists had problems with MSDs; afterwards, it was 81%. Finally, participation in the educational intervention program improved the symptoms of MSDs significantly and reduced MSDs in dentists by teaching good working postures [18]. In another study [14], 49% of dental students reported an improvement of the symptoms of MSDs 3 months after a training course in ergonomics (*p* < 0.05), although 17% reported a worsening of symptoms. The training course showed mutual results but a clear benefit for half of the participants and therefore was an effective option to reduce the symptoms of MSDs in dental students through improving knowledge of prevention strategies against MSDs. Around 25% of the dental students reported more dynamic working postures at follow up, so it can be assumed that the intervention also improved working postures. It was well accepted, as 87.7% of the participants changed their habits in dental work after the training course [14].

## 4. Discussion

This literature review presents the most current state of the research on ergonomic interventions to prevent MSDs or to improve working posture among dental professionals. Our results were drawn from 11 scientific articles published from 2008 to 2018. The literature review revealed five different subjects of ergonomic interventions: ergonomic dental chairs, magnification loupes, prismatic spectacles, dental instruments and training sessions in ergonomics. In all the included studies, the ergonomic interventions had positive impacts on the frequency or severity of MSDs or working posture among dental professionals. This indicates a high level of efficiency and good suitability of the interventions in this context. Therefore, ergonomic interventions can be of importance in dental care and can make a valuable contribution to permanently reducing the prevalence and incidence rates of MSDs among dental professionals. However, as most studies used a rather short follow up time, this conclusion need to be confirmed in further studies. Moreover, ergonomic interventions might improve the ability to work and quality of work, as this was observed in some studies [4,24].

Most of the studies (73%) focused on setting prevention strategies while performing ergonomic interventions. The other studies focused on ergonomic training and behavioral changes. Surprisingly few studies investigate multimethodological approaches combining ergonomic interventions with ergonomic training.

In the studies, the most commonly analyzed body regions were the neck (72.7%, *n* = 8) and the shoulder (72.7%, *n* = 8), which are commonly affected body regions in dentistry. The included studies showed that ergonomic interventions for the neck, shoulder and back regions can be effective.

To different degrees, all the included intervention studies showed positive results either on MSDs or on working posture. Nevertheless, some aspects should be considered when interpreting the individual study results. When conducting intervention studies with an expected positive effect, there is a risk of reporting bias or publication bias in which certain (positive) study results are more likely to be reported or published. Therefore, the results (effects) of the included studies should be interpreted with caution.

Furthermore, the study results belonging to the same subject group were comparable and came to similar conclusions. Dable and colleagues [17] as well as Hallaj and colleagues [19] found out that the use of ergonomic dental chairs with magnification loupes or arm support significantly improved working posture among dental professionals. One other study [26] revealed similar results for surgeons. The use of an ergonomic saddle seat in microsurgery showed significantly better results for physical posture at work compared to the use of conventional seats. This shows that dynamic ergonomic chairs are more suitable for health care work than static chairs. Four included studies [17,20,21,24] reported a decrease in the symptoms of MSDs or an improvement in working posture in dental professionals through the use of magnification loupes in dental care. Other studies confirmed these results. In one study [27], the use of magnification loupes significantly reduced discomfort from MSDs in different body regions such as the neck, shoulder, arm and back among semiconductor assembly workers. Ludwig and colleagues [28] found similar results for dental professionals but without statistical significance. However, follow up surveys indicated that 74% of the participants agreed with wearing magnification loupes as they facilitate dental work and 67% felt that wearing magnification loupes improved their working posture. Furthermore, our review revealed that the use of prismatic spectacles caused positive changes in working posture and reduced the symptoms of MSDs among dental professionals [22,23]. A study from Kuang and colleagues [29] reported comparable results for surgeons in health care. They found that the use of prismatic glasses significantly reduced pronounced neck flexion during cleft palate surgery, and visual analog scale discomfort scores significantly decreased for the neck, back and shoulder regions after the intervention. The use of prismatic spectacles, therefore, is very useful in health professions and suitable to prevent MSDs or to improve working postures. Moreover, one included study [25] revealed that the use of a lightweight dental instrument with a wide diameter can improve the symptoms of MSDs among dental professionals. The study also showed that the weight and diameter of a dental instrument has an influence on the prevalence of MSDs. Further research on this topic should be conducted as there are few studies on this topic. Finally, our work included three studies [14,16,18] that found training courses in ergonomics to improve the symptoms of MSDs or to improve working posture among dental professionals. This showed that knowledge about ergonomics, workplace modification and prevention strategies can contribute to better health. Other studies revealed comparable results. In one study [30], a workplace-based multifaceted intervention including participatory ergonomics was tested to manage MSDs and its consequences for the workers of a medium-sized company. The authors reported that the rates of MSDs (*p* < 0.01) and absenteeism from work (*p* < 0.05) were both significantly reduced after the intervention. Most of the participants agreed that the intervention improved their health status [30]. Similar results were found by Sanaeinasab and colleagues [31] for office computer workers in hospitals. They analyzed the effect of a trans-theoretical model (TTM)-based educational program on work-related posture. The intervention was effective in improving the ergonomic working posture of the computer workers.

In addition, it should be considered that MSDs are a multifactorial problem. Therefore, the effect of a single intervention is limited. In multimethodological approaches, different aspects of the dental work and the characteristics of the study participants should be considered, such as the length of employment and working hours in dentistry, the number of patients, other job-related burdens, resources, age, personal pre-concomitant and concomitant diseases and dispositions, and personal factors in dealing with MSDs (e.g., stress management and attitude). Thus, a positive effect of the ergonomic intervention on MSDs or working posture should not be attributed solely to the intervention itself. Most of the studies included in this review did not analyze the influence of possible confounders on the study outcome after the intervention.

Besides the described ergonomic interventions, several studies showed that other measures such as physical activity (e.g., yoga or fitness courses), physical therapy and complementary and alternative medicine (CAM) therapies can also have a positive impact on the prevalence of MSDs among dental professionals [32,33,34,35,36,37,38]. Therefore, the prevention of MSDs is complex and can be promoted by many different factors and measures in the workplace. As this review showed, ergonomic interventions thereby play a major role and can make a valuable contribution to the prevention of MSDs among dental professionals.

### Strengths and Limitations

This literature review and its included studies contain various methodological strengths and limitations. Firstly, this work considered studies from all over the world, with no geographical restrictions. Therefore, we were able to describe studies from many countries across different continents. This provides a global view of the presented research topic, and different perspectives and subjects are discussed. However, through the global view, the working conditions, work load and environmental factors of the relevant dental care facilities can be quite different and are hard to compare. That is why a reasonable comparison of the included studies is only possible to a limited degree.

In addition to geographical and cultural diversity, other factors make the comparability of the studies difficult, such as the use of various study designs, survey methods and instruments, outcome measures, and subjects of ergonomic intervention. The included studies differ greatly from each other in their methodological approaches. Therefore, it could be difficult to draw general conclusions for this work. Nevertheless, the included studies showed valid results from which recommendations for practical solutions can be derived.

The literature search revealed only a small number of relevant studies (*n* = 11). Considering the high burden of MSDs among dental professionals, this indicates a further need for research. Our review might be useful for conceiving further studies that should focus on multimethodological approaches.

Because of the low comparability and small size of present studies, it was not possible to perform further analyses like stratification, sensitivity or meta-analyses. In addition, not enough usable data were available for risk of bias analyses, but we have considered the risk of bias in our quality assessment according to Downs and Black [15] and evaluated the consideration of a choice of bias for each included study.

A limitation in the study design of seven included studies [14,17,18,19,20,21,23] is that no randomization was used to allocate the participants randomly to the respective investigation groups. In six studies [14,17,18,19,24,25], no real control groups were included to analyze the effect of the ergonomic intervention through a direct comparison between an intervention and control group. In this case, own controls were used instead; thereby, the study participants represent both the intervention and control group at baseline and follow up.

Besides the described limitations, this literature review and its included studies also showed methodological strengths. Firstly, the present literature review was carried out systematically in line with the PRISMA guidelines [11]. The PRISMA guidelines are well accepted, clearly structured and user-friendly for scientists who intend to conduct literature reviews and/or meta-analyses of intervention studies systematically.

The quality assessment of the included studies was performed with a validated standardized instrument [15]. The instrument was created for the assessment of the methodological quality of randomized and non-randomized studies of health care interventions, so it is very suitable for a valid quality assessment in this context. The study quality of the included sources was good, with an average of 19.6 points. Most of the studies (7, 64%) were of moderate quality, but around one third (4, 36%) were of high quality. Positively, there were no studies of low quality. However, the quality assessment revealed some weaknesses in methodology. The most common limitations in the studies were no randomization or blinding used, no control for confounding in statistical analyses and no calculation of power. In some studies, information about losses to follow up was missed.

Furthermore, this review only considered intervention studies published in peer reviewed journals and no grey literature. Therefore, sufficient methodological quality of the studies was ensured so that reliable conclusions could be drawn.

Additionally, the included studies used various survey instruments to evaluate the effects of the respective ergonomic interventions on MSDs or working posture among dental professionals. Standardized and validated questionnaires like the DASH questionnaire or the NPDS scale were most commonly used (72.7%), followed by questionnaires with self-developed questions (63.6%), photographs or videotapes (36.3%), standardized posture assessment instruments like RULA or PAI (27.2%), and physical examinations (27.2%). The original DASH questionnaire is fully validated, is well known in this research field and showed an excellent test–retest reliability (ICC = 0.96) as well as good validity [39]. It seems to be an appropriate tool to analyze the effect of ergonomic interventions on MSDs. The clinical examinations were performed in line with standardized protocols or checklists. Overall, the included studies used suitable survey instruments.

## 5. Conclusions

Several ergonomic interventions to prevent MSDs among dental professionals were found to show positive effects on the prevalence of MSDs or working posture. Our findings revealed five different subjects of ergonomic interventions (ergonomic dental chairs, magnification loupes, prismatic spectacles, ergonomic dental instruments and training courses in ergonomics) that successfully contributed to the reduction in MSDs or the improvement of working posture among dental professionals. This review adds current evidence for the use of prismatic spectacles in order to prevent MSDs. However, as most studies had rather short follow up periods, the long-term effects of these interventions are still to be verified. Further studies are warranted. In accordance with the general discussion in ergonomics, future studies should focus on multimethodological approaches.

## Figures and Tables

**Figure 1 ijerph-17-03482-f001:**
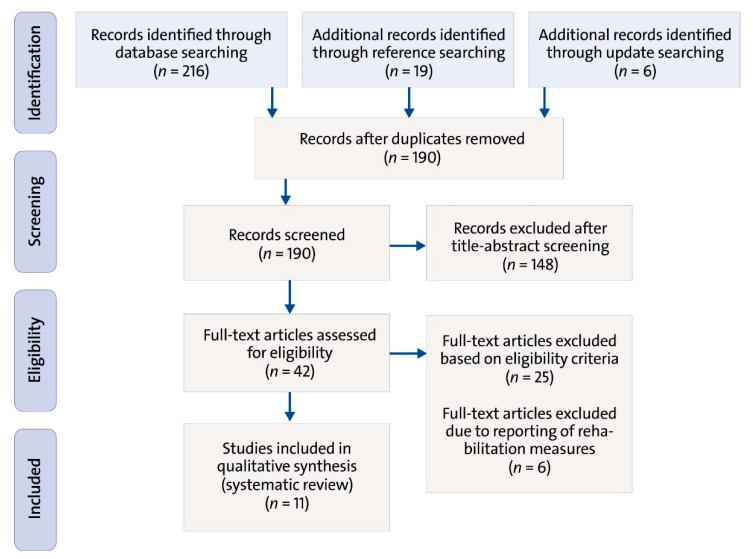
Study selection process for this systematic review (Preferred Reporting Items for Systematic Reviews and Meta-Analyses (PRISMA) Flowchart).

**Figure 2 ijerph-17-03482-f002:**
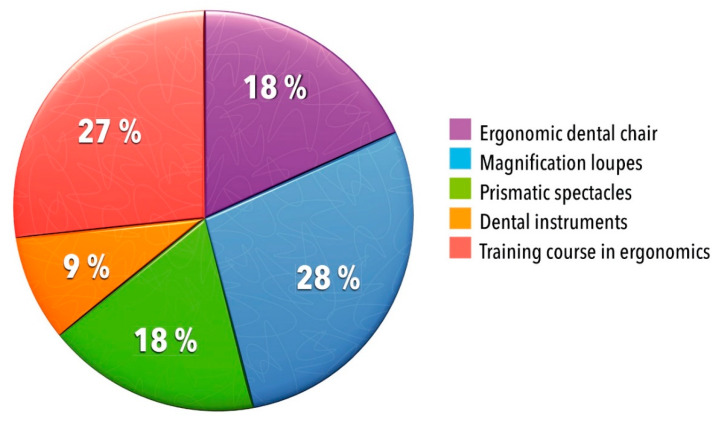
Subjects of ergonomic interventions to prevent MSDs among dental professionals (*n* = 11).

**Table 1 ijerph-17-03482-t001:** Eligibility criteria for the study selection.

PICOS	Study Inclusion Criteria
Population (P)	Dental professionals: e.g., dentists, orthodontists, dental assistants/hygienists/technicians/surgeons/students, dental laboratory assistants
Intervention (I)	Ergonomic interventions that lasted for at least two days
Control group (C)	All suitable control groups, including subjects representing both the intervention and control group (here, own controls)
Outcome (O)	Related to MSDs (prevalence or symptoms) or to working posture
Study design (S)	Intervention or evaluation studies, randomized controlled trials (RCTs), observational studies (e.g., cohort studies), once the effect of the intervention had been clearly analyzed
**Additional Criteria**	
Language	English
Publication status	Published and accessible articles with related abstracts, ideally from peer-reviewed journals
Publication date	January 2008 to May 2018 (update April 2019)

Abbreviations: MSDs: musculoskeletal diseases and pain.

**Table 2 ijerph-17-03482-t002:** Characteristics of the included studies reporting ergonomic interventions to prevent MSDs among dental professionals (*n* = 11).

Reference	Study Design	Country	Setting	Population	Sample Size IG/CG	Related Outcome	Intervention	Study Quality Score
Dable 2014 [17]	Evaluation (between-subject experimental design)	India, Asia	Dental school/university	Dental students	90 see above	Working posture	Ergonomic dental chairs, magnification loupes, lecture in ergonomics	15 (Moderate)
Dehghan 2016 [16]	Intervention (3 survey periods)	Iran, Asia	Dental hospital/clinic	Dentists	102 50/52	Prevalence of MSDs	Training course in ergonomics	21 (Moderate)
Farrokhnia 2018 [18]	Intervention (pre-post intervention design)	Iran, Asia	Dental hospital/clinic	Dentists	84 see above	Symptoms of MSDs	Training course	16 (Moderate)
Hallaj 2016 [19]	Evaluation (pre-post intervention design)	India, Asia	Dental hospital/clinic	Dentists	29 see above	Working posture	Ergonomic dental chair with arm support	11 (Moderate)
Hayes 2014 [20]	Intervention (pre-post intervention design)	Australia, Oceania	Dental school/university	Dental hygienists, dental hygiene students	29 12/17	Symptoms of MSDs	Magnification loupes	18 (Moderate)
Hayes 2016a [21]	Intervention (pre-post intervention design)	Australia, Oceania	Dental school/university	Dental hygienists, dental hygiene students	29 12/17	Symptoms of MSDs	Magnification loupes	23 (High)
Koni 2018 [14]	Intervention (pre-post intervention design)	Italy, Europe	Dental school/university	Dental students	55 see above	Symptoms of MSDs	Training course in ergonomics	22 (High)
Lindegård 2012 [22]	RCT (pre-post intervention design)	Sweden, Europe	Dental hospital/clinic	Dentists, dental hygienists	45 25/20	Working posture	Prismatic spectacles, lecture in ergonomics	23 (High)
Lindegård 2016 [23]	Cohort (longitudinal pre-post intervention design)	Sweden, Europe	Dental hospital/clinic	Dentists, dental hygienists, orthodontic assistants	564 371/193	Symptoms of MSDs	Prismatic spectacles	17 (Moderate)
Maillet 2008 [24]	Intervention (3 survey periods)	Canada, North America	Dental school/university	Dental hygiene students	35 see above	Working posture	Magnification loupes	20 (Moderate)
Rempel 2012 [25]	RCT (pre-post intervention design)	United States, North America	Dental practice	Dentists, dental hygienists	110 54/56	Symptoms of MSDs	Dental instruments	30 (High)

Abbreviations: CG: control group, IG: intervention group, MSDs: musculoskeletal diseases and pain, RCT: randomized controlled trial.

**Table 3 ijerph-17-03482-t003:** Ergonomic interventions and their effects in the included studies reporting ergonomic interventions to prevent MSDs among dental professionals (*n* = 11).

Reference	Type of Prevention	Description of Intervention	Effect of Intervention	Analyzed Body Regions
**Ergonomic Dental Chair**				
Dable 2014 [17]	Setting prevention	Intervention: Ergonomic dental chairs; magnification loupes; lecture in ergonomics. Facts: 3 different dental chairs were analyzed (30 dental students in each group)—(a) saddle stool, (b) conventional chair with back rest, (c) conventional chair without back rest. All investigations on working posture were carried out without and with magnification loupes. All students were lectured on ergonomic posture. After 3 months of training, the assessment procedure started; it lasted for 3 days. Survey instruments: Rapid Upper Limb Assessment (RULA) & videotapes Control: Study participants were their own controls (allocated dental chair without vs. with magnification loupes) Length: 3 months Follow up: In 3 days MSD status of participants: Is not stated, but there is a hint that some participants had mild MSDs before the intervention	- The use of the 3 different dental chairs (without and with magnification loupes) had various effects on working posture of dental students, e.g., The study showed significantly lower RULA scores for the saddle stool with magnification used (1.57 ± 0.50) as compared to the conventional chairs without magnification used (7.03 ± 0.49); for the saddle stool with magnification, the scores were very acceptable (*p* < 0.01) 🡪 The use of the ergonomic saddle stool with magnification loupes was more suitable for dental students and produced a better working posture than the use of the conventional chairs without magnification loupes 🡪 The use of the ergonomic saddle stool and loupes significantly improved the working posture of dental students (*p* < 0.01) 🡪 Dental students reported to have fewer or no MSDs after using the saddle stool as they found it more comfortable to work in this chair than in the conventional chairs - The use of magnification loupes influenced the working posture of dental students for every dental chair, e.g., The study reported significantly lower RULA scores for the conventional chairs with magnification (CC1 5.63 ± 0.49 and CC2 5.07 ± 0.46) than in the groups without it (6.57 ± 0.50 and 6.96 ± 0.56) 🡪 The use of magnification loupes significantly improved the working posture of dental students (*p* < 0.01) - When the conventional chairs were compared, it was seen that the back rest does not make any difference in improving the working posture of dental students (*p* > 0.05) - The study reported that the use of the ergonomic saddle stool could support the lumbar region and maintain the natural curvature of the lower back; at the same time, magnification could bring a clearer view near to the operator instead of the operator hunching over to get the view	- Arm - Leg - Neck - Trunk - Wrist
Hallaj 2016 [19]	Setting prevention	Intervention: Ergonomic dental chair with arm support Facts: A new designed arm support device was tested Survey instruments: Rapid Upper Limb Assessment (RULA) and photographs, feedback questionnaire with self-developed questions Control: Study participants were their own controls Length: 1 week Follow up: Time is not stated MSDs status of participants: Is not stated	- The use of an ergonomic dental chair with arm support correlated with the working posture of dentists, e.g., the overall RULA score (average value of all participants) was 3.14 after the use of an ergonomic dental chair with arm support 🡪 The use of an ergonomic dental chair with arm support led to positive changes in the working posture of dentists - The use of an ergonomic dental chair with arm support had the following effects on the working posture of dentists, e.g., the combined bending and twisting of the back decreased by 13.8% after using the arm support device; the twisting, turning, grapping and wringing actions with fingers or arms bent decreased by 20.7%; excessive bending up or down of the wrist decreased by 41.38%; pinch grip decreased by 17.2%; the pressure on the neck and shoulder while performing dental tasks decreased by 79.3% 🡪 The use of an ergonomic dental chair with arm support had a significant positive impact on the working posture of dentists 🡪 Dentists reported more comfort by using the arm support device - The working posture can further be improved by adjusting both the patient’s and dentist’s chairs, to support the dentist’s neck during work - The RULA score indicated that by using the arm support device, the body posture of dentists is almost in the correct ergonomic position - Dentists stated that they prefer to have only one side arm supported	- Arm - Back - Elbow - Head - Neck - Shoulder - Wrist
**Magnification Loupes**				
Hayes 2014 [20]	Setting prevention	Intervention: Magnification loupes Facts: Galilean flip-up style loupes with 2.5 x magnification were used. The convergence and working angles of the magnification loupes were adjustable. Survey instruments: Disabilities of the Arm, Shoulder and Hand (DASH) Questionnaire, physical assessments based on validated protocols Control: Dental hygiene students not wearing magnification loupes Length: 6 months Follow up: After 6 monthsMSDs status of participants: All study subjects experienced MSDs before the intervention, subjects with chronic MSD conditions were not included in the study	- The use of magnification loupes in dental care was associated with symptoms of MSDs among dental hygienists, e.g., DASH scores for dental hygienists (intervention group) were higher than for dental hygiene students (control group) (8.56 ± 9.64 vs. 4.99 ± 6.25) at baseline; this trend reversed after the intervention (5.17 ± 5.29 vs. 7.84 ± 8.73) 🡪 Following the intervention, the DASH scores for dental hygienists decreased, and those for dental hygiene students increased 🡪 The use of magnification loupes significantly reduced symptoms of MSDs among dental hygienists (p < 0.05) - Levels of self-reported upper extremity pain and disability improved in the intervention group when comparing baseline to post-intervention, while symptoms of MSDs in upper extremities worsened in the control group - Changes in musculoskeletal function were minimal among dental hygienists - Dental hygienists reported less pain in the shoulder, arm and hand regions after the intervention	- Arm - Hand - Shoulder
Hayes 2016a [21]	Setting prevention	Intervention: Magnification loupes Facts: Galilean flip-up style loupes with 2.5 × magnification were used. The convergence and working angles of the magnification loupes were adjustable. Survey instruments: Neck Pain and Disability Scale (NPDS), physical assessments based on validated protocols Control: Dental hygiene students not wearing magnification loupes Length: 6 months Follow up: After 6 months MSDs status of participants: All study subjects experienced MSDs before the intervention, subjects with chronic MSDs conditions (persistent pain for at least 3 months) or with pre-existing MSDs unrelated to occupational factors were not included in the study	- The use of magnification loupes correlated with neck pain and disability in dental hygienists, e.g., The study revealed no significant interactions between time and treatment (*p* > 0.05); there was no change in mean NPDS scores between baseline and follow up for the intervention group (14.00 ± 12.49 vs. 14.00 ± 11.05), while the control group reported an increase in perceived neck pain at follow up (14.97 ± 16.91 vs. 15.90 ± 13.54) (*p* > 0.05) 🡪 The use of magnification loupes created no significant changes in neck pain and disability in dental hygienists over time 🡪 The use of magnification loupes had no significant effect on improving symptoms of neck pain and disability in dental hygienists, but a slightly positive impact can be assumed	- Neck
Maillet 2008 [24]	Setting prevention	Intervention: Magnification loupes Facts: The magnification loupes were Hires flip-ups, complete with head straps and side shields. The frames were all standard titanium frames, slate in color. Orascoptic also provided three rigid headbands to allow for prescription eyeglass wearers. The headbands and standard frames had interchangeable working lengths to allow for portability within the group. The magnification for all was 2.5 ×. The study consisted of two parts: preliminary study and formal study that were implemented in 2005. Group 1 wore the loupes for the first session and worked without them for the second session, while Group 2 worked without loupes for the first session and with loupes for the second. Survey instruments: Posture Assessment Instrument (PAI), Posture Assessment Criteria (PAC), post-study-survey with self-developed questions and videotapes Control: Study participants were their own controls (2 sessions each with and without magnification loupes) Length: 7 months Follow up: After 7 monthsMSDs status of participants: Is not stated	- The use of magnification loupes showed effects on the working posture of dental hygiene students, e.g., The results of the first session indicated that Group 1 (wore the magnification loupes) had significantly better ergonomic scores than Group 2 (did not wear the magnification loupes). Group 1 had a mean score of 5.69 ± 2.17 points from the ideal posture, compared with a mean score of 10.76 ± 4.30 points for Group 2 (t = 4.37, df = 23, *p* < 0.001); in the second session, Group 2 (wore the magnification loupes) had significantly better ergonomic scores than Group 1 (did not wear the magnification loupes). Group 2 had a mean score of 7.83 ± n/a points from ideal posture, compared with a mean score of 10.13 ± n/a points for Group 1; in the end, all students wearing magnification loupes showed significantly better ergonomic scores than all students not wearing magnification loupes (6.4 ± 2.61 vs. 10.8 ± 4.24, t = 6.66, df = 34, *p* < 0.000001) 🡪 The use of magnification loupes significantly improved the working posture of dental hygiene students in both groups (*p* < 0.001) 🡪 An early introduction in magnification loupes was more effective in improving the working posture - The majority of students were aware of the improved posture, perceived that the quality of their work increased when wearing magnification loupes and would wear loupes regularly if they were provided	- Arm - Head - Hip - Leg - Neck - Shoulder - Trunk
**Prismatic Spectacles**				
Lindegård 2012 [22]	Setting prevention	Intervention: Prismatic spectacles; lecture in ergonomics Facts: The prismatic glasses include optometric correction. The ergonomic education (lecture in ergonomics) includes a comprehensive 1.5 h information session about dental ergonomics including working postures, working technique and visual ergonomics. All study participants underwent the education. The assessments lasted 4 months. Survey instruments: Borg’s RPE Scale (modified), inclinometers and questionnaires Control: Dentists and dental hygienists not wearing prismatic spectacles Length: 12 months Follow up: CG: 7 and 8 weeks after the education, IG: 9 to 11 weeks and 12 months after the intervention MSD status of participants: Is not stated	- The use of prismatic glasses in dental care had an impact on the working posture of dentists and dental hygienists, e.g., at follow up, the head flexion was reduced in both groups but more pronounced in the intervention group (received prismatic glasses) than in the control group (did not receive prismatic glasses) (8.7° vs. 3.6°, *p* < 0.01); regarding the neck flexion, a significant reduction was seen for the intervention group, while a smaller and insignificant reduction was present in the control group (8.2° vs. 3.3°, *p* < 0.05); in the intervention group, there was a significant decrease (4 units) in the perceived exertion of the head and the neck at follow up, and the corresponding decrease for the control group was 2 units (n. s.) 🡪 The use of prismatic glasses made significant positive changes in the working posture of dentists and dental hygienists for the head and the neck regions 🡪 The use of prismatic glasses reduced complaints in the head and the neck caused by dental work 🡪 The use of prismatic glasses facilitated the performance of dental work (🡪 80% of the participants reported that the prismatic glasses were feasible to wear during work and considerably facilitated dental work) 🡪 The use of prismatic glasses decreased the risk of exposure to high risk working postures in the neck during dental work	- Head - Neck
Lindegård 2016 [23]	Setting prevention	Intervention: Prismatic spectacles Facts: All participants in the intervention group were given an eye test for adjusting the prismatic glasses individually. Survey instruments: Nordic Musculoskeletal Questionnaire (NMQ), Work Ability Index (WAI), questionnaire with self-developed questions for the follow up assessment, physical assessments based on Health Surveillance in Adverse Ergonomics Conditions (HECO) protocols Control: Remaining dental personnel not receiving prismatic spectacles Length: 12 months Follow up: After 12 months MSE status of participants: All study subjects experienced MSDs before the intervention (at baseline); the intervention group reported a higher prevalence of MSDs and clinical diagnoses at baseline than the control group	- The use of prismatic glasses during clinical dental work correlated with symptoms of MSDs in dental personnel, e.g., the study revealed in the intervention group (received prismatic glasses) significant improvements regarding clinical diagnoses (*p* < 0.05), perceived exertion (*p* < 0.01), self-reported pain (*p* < 0.05) and self-rated work ability (*p* < 0.05) compared to the control group (did not receive prismatic glasses) 🡪 The use of prismatic glasses significantly improved symptoms of neck and/or shoulder pain in dental personnel 🡪 The use of prismatic glasses significantly reduced the risk of developing MSDs (including neck and shoulder pain) and decreased perceived muscular exertion during the performance of dental work - The prismatic glasses enable the dental personnel to work in a more upright position with a less bent neck that promotes an ergonomic working posture with a lower risk of developing muscular complaints and symptoms of MSDs - Study participants reported that wearing the prismatic glasses simplified their daily work and strengthened their work ability in dental care 🡪 The greatest advantage of the prismatic glasses was found during root-fillings and other vision-demanding tasks in constrained working positions	- Neck - Shoulder
**Dental Instruments**				
Rempel 2012 [25]	Setting prevention	Intervention: Dental instruments Facts: Instrument 1 weighed 14g and had an 11mm diameter handle, Instrument 2 weighed 34g and had an 8mm diameter handle. Instrument 1 was made from black plastic, and Instrument 2, from steel plated with black coating. Randomization took place at the level of the dental office. Survey instruments: Online questionnaires at baseline, weekly during the intervention and at follow up Control: 2 intervention groups with own controls (use of light/wide vs. heavy/narrow instrument) Length: 5 months Follow up: After 5 months MSDs status of participants: Study subjects experienced MSDs before the intervention; subjects who received any treatment of MSDs before the intervention were not included in the study	- The use of a lightweight dental instrument with a wide diameter had impacts on symptoms of MSDs in dentists and dental hygienists, e.g., the unadjusted pain scores improved more for study participants who used Instrument 1 (light and wide) than for those who used Instrument 2 (heavy and narrow) for the wrist/hand (0.40 ± 0.11 vs. 0.14 ± 0.11, n. s.), arm (0.20 ± 0.09 vs. 0.06 ± 0.09, n. s.) and shoulder (0.51 ± 0.16 vs. 0.19 ± 0.15, *p* < 0.05) regions; after adjusting for confounders (e.g., age and occupation), the authors found a significant difference between the two groups only for the shoulder region (0.52 ± 0.17 vs. 0.19 ± 0.16, *p* < 0.05) 🡪 The use of the lightweight dental instrument with a wide diameter significantly reduced symptoms of shoulder pain in dentists and dental hygienists 🡪 The improvements in symptoms of MSDs were greater among those who used the lightweight instrument with a wide diameter 🡪 The use of the lightweight instrument with a wide diameter was more suitable for dental work than the use of a heavyweight instrument with a narrow diameter, even if symptoms of MSDs improved in both groups - The number of nights awakened with finger numbness improved more for participants assigned to the lightweight instrument with a wide diameter than they did for those assigned to the heavyweight instrument with a narrow diameter - The follow up survey ratings regarding the usability of the instruments were more positive for participants who used the lightweight instrument with a wide diameter than they were for those who used the heavyweight instrument with a narrow diameter	- Arm - Hand - Shoulder - Wrist
**Training Course in Ergonomics**				
Dehghan 2016 [16]	Behavioral prevention	Intervention: Training course in ergonomics Facts: The intervention includes 4 sections: 1. knowledge and training about ergonomics (training sessions), 2. workstation modification (instructions how to modify working postures), 3. training and surveying ergonomics at the workstation (working conditions were evaluated, discussed and modified), 4. regular exercise program (stretching movements were explained by a physiotherapist). Survey instrument: Nordic Musculoskeletal Questionnaire (NMQ) Control: Dentists not receiving the ergonomic intervention program Length: 2 months Follow ups: After 3 and 6 months MSD status of participants: Is not stated	- Participation in the ergonomic intervention program influenced the prevalence of MSDs in dentists, e.g., dentists who were in the intervention group had lower prevalence rates of MSDs for all surveyed body regions at 3 and 6 months after the program than dentists who were in the control group; e.g., the prevalence of knee pain was 24% in the intervention group and 36% in the control group 6 months after the program (*p* < 0.01); the prevalence of shoulder pain was 44% and 80% (*p* < 0.05), and of neck pain, 62% and 84% (*p* < 0.01); prevalence rates of MSDs decreased over time in the intervention group for all body regions and in the control group only for the back region; e.g., in the intervention group, the prevalence of knee pain was 30% before and 24% 6 months after the program (*p* < 0.01); the prevalence of shoulder pain was 60% and 44% (*p* < 0.01), and of neck pain, 78% and 62% (*p* < 0.01); therefore, prevalence rates of MSDs increased over time in the control group for almost all body regions 🡪 The ergonomic intervention program had a positive effect by significantly reducing the prevalence of MSDs in dentists 🡪 Theoretical and practical knowledge about ergonomics and workplace modification in dental care can significantly improve the experience of MSDs in dentists - Almost all surveyed dentists (98%) who were in the intervention group agreed with the multifaceted ergonomic intervention program and experienced a positive benefit, finally had significantly fewer MSDs after the intervention and were able to improve their workplace ergonomics through gained knowledge	- Arm - Back - Foot - Knee - Neck - Shoulder - Thigh - Wrist
Farrokhnia 2018 [18]	Behavioral prevention	Intervention: Training course Facts: The educational intervention included a brief face-to-face teaching and distributing pamphlets Survey instruments: Cornell Musculoskeletal Discomfort Questionnaire (CMDQ) and questionnaire with self-developed questions Control: Study participants were their own controls Length: More than two days (probably a few weeks) Follow up: After 2 months MSDs status of participants: Most of the study subjects (87%) experienced MSDs before the intervention; some study subjects (13%) were free of MSDs at this time	- Participation in the educational intervention correlated with the symptoms of MSDs in dentists, e.g., at follow up the study results revealed a significant reduction in means for MSDs for the neck (10.97 ± 20.44 vs. 7.91 ± 17.01, *p* < 0.01), right shoulder (8.85 ± 19.76 vs. 5.24 ± 13.51, *p* < 0.01), left shoulder (5.80 ± 17.21 vs. 2.95 ± 9.33, *p* < 0.01), upper back (6.92 ± 17.59 vs. 4.53 ± 14.35, *p* < 0.01) and right wrist (5.12 ± 13.35 vs. 3.81 ± 12.96, *p* < 0.05) regions; before the intervention, 87% of dentists had problems with MSDs; after the intervention, it was 81% 🡪 Through participation in the educational intervention, symptoms of MSDs significantly improved in dentists 🡪 The educational intervention had the greatest impact on body regions like the neck, shoulder, back and wrist 🡪 The educational intervention had positive effects on present symptoms of MSDs and contributed to reducing MSDs in dentists by teaching good working postures, regular rest breaks and stretching exercises - Further analyses showed that more short breaks between patients resulted in lower MSDs (*p* < 0.05) and increased age led to more neck pain (*p* < 0.05)	- Arm - Back - Hip - Knee - Leg - Neck - Shoulder - Thigh - Wrist
Koni 2018 [14]	Behavioral prevention	Intervention: Training course in ergonomics Facts: The intervention comprised several training sessions, each of 60 minutes in length. The program was organized by the University of Trieste, School of Dentistry and Physiotherapy degree course. The course taught the participants in basic knowledge on working postures and MSDs and in prevention strategies against symptoms of MSDs. Survey instruments: Verbal Numerical Scale (VNS), photographs and questionnaires Control: Study participants were their own controls Length: More than two days (probably a few weeks) Follow up: After 3 months MSDs status of participants: Is not stated, but there is a hint that all study participants had some form of MSDs before the intervention	- Participation in the training course in ergonomics was associated with symptoms of MSDs in dental students, e.g., 49% of dental students reported an improvement of symptoms of MSDs 3 months after the training course (*p* < 0.05), but 17% reported a worsening of symptoms; women, younger students and those who reported less pain at the beginning of the study experienced fewer improvements of symptoms of MSDs after the intervention (OR = 0.48, 95% CI = 0.22-1,04; OR = 0.93, 95% CI = 0.83–1.03; OR = 0.94, 95% CI = 0.89–0.99) 🡪 The training course showed mutual results, but a clear benefit for half of the surveyed dental students can be derived 🡪 The training course is an effective option to reduce symptoms of MSDs in dental students through improving knowledge of prevention strategies - Of the dental students, 25.6% reported more dynamic working postures at follow up 🡪 The training course effectively improved working postures in dental students - Of the dental students, 87.7% changed their habits in dental work after the training course following its suggestions for a better working posture and prevention strategies against MSDs 🡪 The training course was well accepted and provided practical skills for dental students	- Back - Elbow - Foot - Hand - Head - Hip - Knee - Shoulder

Abbreviations: CC1: conventional chair 1; CC2: conventional chair 2; CG: control group; CI: confidence interval; DASH: Disabilities of the Arm, Shoulder and Hand; df: degrees of freedom; IG: intervention group; MSDs: musculoskeletal diseases and pain; NPDS: Neck Pain and Disability Scale; n/a: not applicable; n. s.: not significant; OR: odds ratio; RPE: Received Perception of Exertion; RULA: Rapid Upper Limb Assessment.

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
