# Peer review of "Prevention of Musculoskeletal Diseases and Pain among Dental Professionals through Ergonomic Interventions: A Systematic Literature Review"

_ijerph, 2020, doi:10.3390/ijerph17103482_

Round 1

Reviewer 1 Report

The authors present a review article on MSD' and related intervention in dental profession. At the end they conclude based on this review that such interventions are beneficial in reducing MSD's in dental workplace. The review is very well done and described . However there are few major concerns to address before this article is suitable for publication. Hope thee comments are useful.

(1) The authors have to clearly make a case at the end of introduction as to ( lines 53:55) :  (a) How does this review differ from others they have listed. What is the need for this review if there are already some? There has to be a rationale for publishing this review. What value does this add to literature which other reviewers did not or hoe does this review complement previous such efforts?

(2) Did the authors explore any cohort P specific  classification/trends for  interventions or injury sites (knee, neck, back, etc) . For example, did the manuscripts on students P end up having a certain kind of intervention types more effective/ frequented injury sites  or used a specific intervention more frequently Vs other P group.? Perhaps a representation like in Figure 2 for this or some text discussion on this if relevant would be beneficial for the reader.

(3)  MSD's can also occur due to repeated overuse in addition to ergonomic factors related to work setup. This review is purely based on look at the extrinsic factors like Dental chair etc. Older individuals have more risk to get exposed to MSD's ( lower threshold for repeated strain/stress work) than younger individuals in general due to aging related muscle and physiologic changes. Since the sample studied (P) has a wide spectrum of age group from student to experienced professional (who are perhaps older). Did the authors find any trends between intervention type, injury sites or effects with respect to age?  I highly recommend the authors to acknowledge that this review  has not looked into workload related factors and age impacts to acknowledge the fact that this is a multifaceted problem and just a intervention might alone not be a total solution.

(4) Line 365:367: s this statement authors speculation. Since it seems like an affirmative statement, please add supportive references from literature for this statement

(5) Did all these 11 studies have any report of pre-condition for MSD's before study intervention? Were all the participants free off/ or had MSD during thee studies? Any details from these 11 studies on inclusion and exclusion criteria for each will be beneficial to include as a summary.

Conclusion:

One other aspect the authors did not consider is the inflow of patient numbers. MSD's can also be influenced by the number of patients each profession attends to every day , which varies depending on the geopolitical location. Although the authors claim they have included reearch work from many parts of the works, the authors have to loop in this context of geopolitical related work load into the discussion of MSD and provide their perspective on that and what this review can or cannot offer towards that end.

Minor english language:

Please check line numbers 190, 185 and 46 for grammatical or typo errors.

All the best to the authors.

Reviewer 2 Report

General comments
1. This study reports a systematic review of the literature on the prevention of musculoskeletal diseases and pain in dental professionals. The study is generally well-written, with a clear background and study aim. The study is reported following the PRISMA guidelines, ensuring most of the required information is present. The results are clearly presented and are in context of the study design. Discussion and Conclusion sections are not strongly consistent with the presented data and requires an extensive review. Among minor suggestions, I have two major concerns related to the rationale of the study and inclusion criteria that I would like the authors to revise or rebut. Please see below.

Specific comments
1. Abstract. The last two sentences (conclusions?) could be rephrased to show what this study adds in terms of the current evidence (and its quality) and whether more studies are necessary.

2. Keywords. Most keywords are not indexed in MeSH (https://www.ncbi.nlm.nih.gov/mesh): dental professionals; prevention; musculoskeletal diseases and pain; systematic review; PRISMA. Consider replacing them (e.g. systematic review as a topic).

3. Introduction. As mentioned, systematic reviews are already available [8-10], including a recent one published in 2019 [10]. Please consider presenting a technical rationale (e.g. quality of evidence, number of studies) for updating the last published review. For instance, so far it is unclear whether your study included papers not yet included in those previous reviews. It is particularly important because most studies were published in 2016 and the most recent study was published in 2018.

4. Eligibility criteria. An inclusion criterion was studies in which ‘improved working postures resulting in a reduction of MSDs were the result of the ergonomic intervention’. However, it is not explicit if you also included studies with the other outcomes (no change or increased MSD as a results of the ergonomic intervention). Nonetheless, from Table 2 it is apparent the case. I am interested in how many studies were not included for no compliance to this inclusion criteria as this could lead to a bias in your findings.

5. Ergonomic interventions. The sentence ‘Almost all results were statistically significant’ is of little value because effect sizes (that are more interesting form the clinical point-of-view) were not reported nor meta-analyzed. Of greater concern is the fact that all published studies reported ‘positive’ effects, what is odd in medical sciences in general – and is probably linked to a publication bias or study selection bias for this review.

6. Prismatic Spectacles (and Training Course in Ergonomics). Please be consistent when reporting data from previous studies. Some sections (Ergonomic Dental Chair, Magnification Loupes, Dental Instruments) report mean and SD values. Other sections (Prismatic Spectacles, Training Course in Ergonomics) repot mean values only.

7. Discussion. Please double check the first paragraph of the Discussion as the sentence ‘Almost all results were statistically significant’ is repeated there. It also repeats the argument that all studies showed positive results, which must be followed by a strong discussion on this subject.

8. Discussion. I agree that no ‘geographical restriction’ was imposed to the study, but by including studies only published in English this somehow limits the participation of research developed by researchers from non-English language countries.

9. Conclusions. Consider being more concise in this section. Also, limit your statements for the available information; it seems that analysis of the  burden of MSD was not even assessed but is stated as a conclusion.

Round 2

Reviewer 1 Report

Towards Conclusion: the authors mention that the research from various counties are included.recommend acknowledging that the aspect of work load Based on county location. was not included in this review.

The authors have addressed all issue other in the responses.

Reviewer 2 Report

Thank you for submitting a revision and for discussing your report. All my previous concerns were sufficiently addressed. I have no new comments.